# Anderson transition in stoichiometric Fe$_2$VAl: high thermoelectric performance from impurity bands

Fabian Garmroudi [1✉], Michael Parzer [1], Alexander Riss [1], Andrei V. Ruban[2,3], Sergii Khmelevskyi [4✉], Michele Reticcioli [5], Matthias Knopf[1], Herwig Michor [1], Andrej Pustogow [1], Takao Mori [6,7] & Ernst Bauer[1]

Discovered more than 200 years ago in 1821, thermoelectricity is nowadays of global interest as it enables direct interconversion of thermal and electrical energy via the Seebeck/Peltier effect. In their seminal work, Mahan and Sofo mathematically derived the conditions for 'the best thermoelectric'—a delta-distribution-shaped electronic transport function, where charge carriers contribute to transport only in an infinitely narrow energy interval. So far, however, only approximations to this concept were expected to exist in nature. Here, we propose the Anderson transition in a narrow impurity band as a physical realisation of this seemingly unrealisable scenario. An innovative approach of continuous disorder tuning allows us to drive the Anderson transition within a single sample: variable amounts of antisite defects are introduced in a controlled fashion by thermal quenching from high temperatures. Consequently, we obtain a significant enhancement and dramatic change of the thermoelectric properties from p-type to n-type in stoichiometric Fe$_2$VAl, which we assign to a narrow region of delocalised electrons in the energy spectrum near the Fermi energy. Based on our electronic transport and magnetisation experiments, supported by Monte-Carlo and density functional theory calculations, we present a novel strategy to enhance the performance of thermoelectric materials.

[1] Institute of Solid State Physics, TU Wien, Vienna, Austria. [2] Department of Materials Science and Engineering, KTH Royal Institute of Technology, Stockholm, Sweden. [3] Materials Center Leoben Forschung GmbH, Leoben, Austria. [4] Center for Computational Materials Science and Engineering, TU Wien, Vienna, Austria. [5] Faculty of Physics, Center for Computational Materials Science, Universität Wien, Vienna, Austria. [6] International Center for Materials Nanoarchitectonics (WPI-MANA), National Institute for Materials Science, Tsukuba, Japan. [7] Graduate School of Pure and Applied Sciences, University of Tsukuba, Tsukuba, Japan. ✉email: fabian.garmroudi@tuwien.ac.at; sk@iap.tuwien.ac.at

Thermoelectric (TE) devices are capable of converting waste heat into useful electrical energy or act as Peltier coolers. Facing an increasing worldwide demand for efficient energy utilisation, the immense diversity of potential technological applications has sparked great interest[1,2]. Still, TE devices are currently restrained in their applicability due to their limited efficiency. The dimensionless figure of merit $ZT = S^2 \sigma T / (\kappa_e + \kappa_{ph})$, which is closely related to the conversion efficiency, comprises three material-dependent parameters. These are the thermopower $S$, the electrical conductivity $\sigma$ and the thermal conductivity $\kappa$, consisting of a contribution from electrons $\kappa_e$ and phonons $\kappa_{ph}$. While considerable progress towards achieving high $ZT$ has been achieved so far by reducing $\kappa_{ph}$[3–10], increasing the electronic part of $ZT$ is a much more formidable, yet necessary task and new exotic concepts for enhancement are required. In 1996, Mahan and Sofo mathematically identified 'the best thermoelectric' as an ideal system, characterised by a delta-distribution-shaped transport function $\Sigma(E)$, where charge carriers are confined in an infinitely narrow energy interval[11].

Here, we propose that this intriguing mathematical concept becomes actually realised in real materials at the Anderson transition in an impurity band, as has been recently discussed theoretically[12]. As sketched in Fig. 1, such a transition occurs when the number of randomly distributed impurities increases above a critical value $x_c$, known as quantum percolation threshold[13]. Below $x_c$, all impurity states are Anderson-localised due to disorder[14]. A singularity of the transport function occurs at $x_c$ when an infinitesimally small region in the density of states (DOS) becomes delocalised. This was explained by Mott in 1967 through the concept of 'mobility edges', which are two critical energies $E_{c_{1,2}}$ that appear at the centre of an impurity band, separating localised states in the band tails from delocalised, extended states in the centre[15]. Far above $x_c$, $E_{c_1}$ and $E_{c_2}$ shift towards the band edges, eventually delocalising all impurity states.

The merit of Anderson localisation to enhance thermoelectricity has been recently shown for the case of one mobility edge, where the electronic states near the valence or conduction band edge of a narrow-gap semiconductor are localised in the presence of a random potential, which may suppress bipolar conduction at higher temperatures[16,17]. Agne et al. argued that simultaneously tuning carrier concentration and disorder for one band with a single mobility edge can increase $ZT$ by $\approx 20\%$[17]. Our study, on the other hand, is focused on the insulator–metal transition in an impurity band (e.g. occurring in lightly doped semiconductors), which has long been considered a fundamental problem in condensed matter physics. In this case, the appearance of two mobility edges directly accomplishes Mahan and Sofo's idea of confined electronic transport (see Fig. 1). We experimentally realised such a scenario in an undoped, stoichiometric bulk $Fe_2VAl$ specimen by controlling the degree of lattice disorder directly via thermal quenching. This Heusler compound was recently found to be an excellent candidate for studying new TE optimisation strategies[18–21]. Our measurements of the electronic transport and magnetisation in this work, supported by Monte Carlo and density functional theory (DFT) simulations, show clear evidence for a significant enhancement of the TE performance, which we attribute to the Anderson insulator–metal transition. In the following, we describe the structural, electronic and magnetic properties of disorder-tuned $Fe_2VAl$ as obtained by our experiments and simulations. Finally, we show the transport properties of the material across the Anderson transition.

## Results

**Structural and electronic properties.** Ternary $Fe_2VAl$ forms a fully ordered $L2_1$ structure at low temperatures that undergoes two second-order structural phase transitions (see Fig. 2a) into the partly disordered B2 structure at $T_{B2} \approx 1100\,°C$ and fully disordered A2 structure at $T_{A2} \approx 1250\,°C$[22]. Our Monte Carlo simulations based on effective cluster interactions show how the degree of atomic disorder in bulk $Fe_2VAl$ can be controlled by temperature, finding a remarkable agreement with the experimental $L2_1$–B2 transition temperature[22] (the B2–A2 transition temperature was overestimated as discussed in "Methods"). This allows us to semiquantitatively assess the concentration of Fe, V and Al atoms on the respective sublattices as depicted in Fig. 2a. Note that, while the V/Al sublattice is fully disordered in the B2 phase, there is already a significant site exchange on the Fe sublattice (5–25% Fe antisites). The large amount of antisite defects obtained at high temperatures as a result of the thermal excitations can be partly frozen by ultrafast quenching our samples.

To showcase the occurrence of narrow impurity states near the Fermi energy $E_F$ in this system, such as those sketched in Fig. 1, we calculated the spin-polarised DOS of Fe/V antisite defects. By making use of the exact muffin-tin orbital coherent potential approximation method (EMTO-CPA) we are able to calculate the DOS of a single impurity embedded in an infinitely large and ordered effective medium, mimicking the electronic and structural properties of an alloy in the dilute limit of antisite concentration $x_{AS} \rightarrow 0$. Figure 2c shows the appearance of sharp, hydrogen-like impurity states near $E_F$ for $Fe_V$ antisite defects, as compared to the fully ordered compound (see Fig. 2b). Similar results are obtained for $V_{Fe}$ and $Fe_{Al}$ defects (see Supplementary Fig. 2). Furthermore, the spin degeneracy is removed due to the strong correlation of Fe-$d$ electrons, which leads to isolated magnetic impurities in the nonmagnetic, ordered host matrix. With increasing quenching temperature and thus increasing antisite concentration, the randomly distributed isolated defects form a continuum of clusters of different sizes[23,24], leading to a

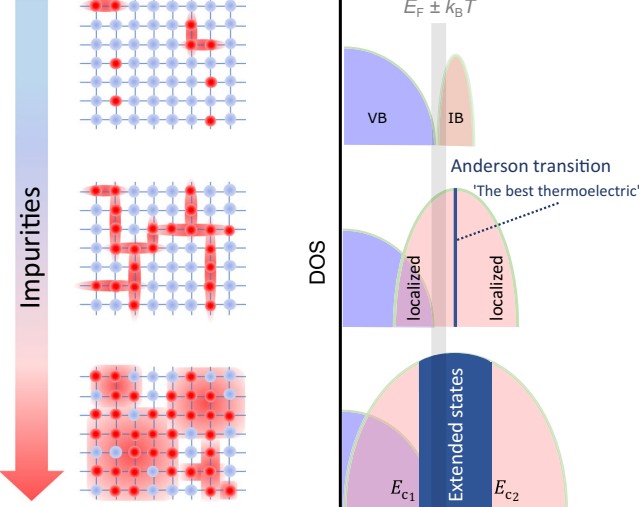

**Fig. 1 Sketch of Anderson transition in an impurity band.** When the number of randomly distributed impurities in a periodocally ordered crystal increases, the impurity electrons remain localised below a threshold value due to Anderson localisation. At the Anderson transition, the critical density of impurities allows for delocalisation of an infinitely narrow energy region of extended states inside the localised impurity states. The delocalised impurity band is marked by two mobility edges $E_{c_{1,2}}$, which are critical energies that separate the localised from delocalised states.

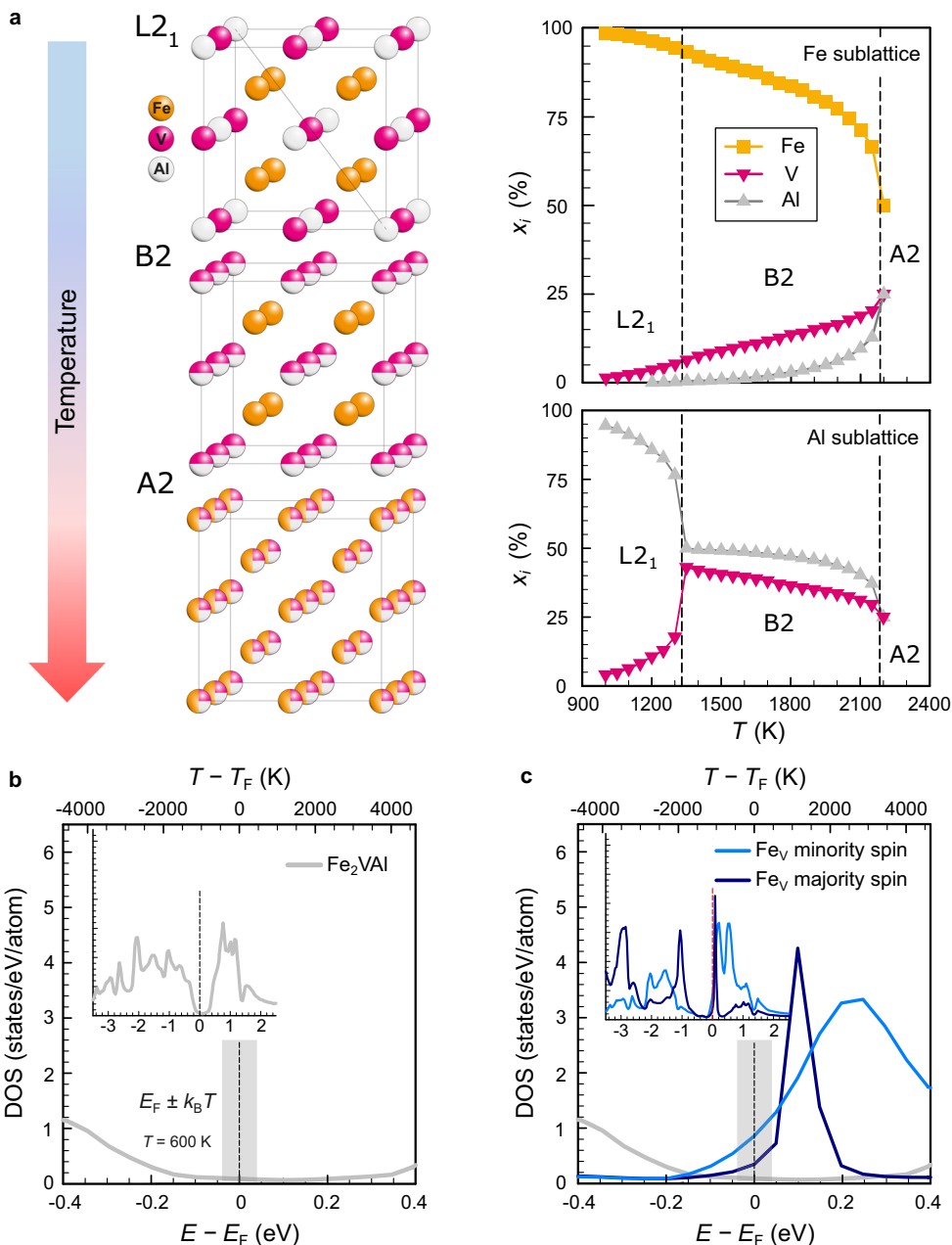

**Fig. 2 Temperature-induced modifications of the electronic structure in Fe₂VAl due to antisite defects. a** Order–disorder transitions and Monte Carlo-simulated concentrations of site occupancies in the L2₁, B2 and A2 high-temperature phases of Fe₂VAl. **b, c** Spin-polarised electronic density of states (DOS) for pure Fe₂VAl and single-impurity Fe$_V$ antisite defects, respectively.

broadening of the localised electronic states (see Supplementary Fig. 3). Eventually, a delocalised impurity band forms, i.e., the Anderson transition, as sketched in Fig. 1. However, neither the CPA nor the supercell approach can determine the critical concentration of defects for the Anderson delocalisation transition[25]. To overcome this difficulty, more effortful methods like the calculation of the typical local density of states[25] could be used as a means of identifying such delocalisation transitions in future works.

**Magnetic properties.** The formation of magnetic clusters predicted by our simulations shown in the previous section (for details see "Methods") can be confirmed by our magnetisation measurements. Figure 3a shows the field-dependent magnetisation $M$ at $T = 4$ K for stoichiometric Fe₂VAl, heat-treated at

different conditions. Measurements of the magnetisation have previously shown to be an effective way of probing Anderson-localised states in other semiconductors such as Si[26]. The immediate saturation of $M$ at small fields observed in Fig. 3a, the absence of hysteresis as well as the strong curvature of isothermal Arrot plots (see Supplementary Fig. 7) are strong indications that the magnetic properties are dominated by the magnetic moments of the randomly distributed antisite defects, in line with our ab initio calculations (see Supplementary Fig. 4). In Fig. 3b, we compare the saturation magnetisation $M_{sat}$ of our samples with their quenching temperature $T_{quench}$. It can be clearly seen that $M_{sat}$ consistently increases for higher $T_{quench}$, corroborating the picture drawn by our simulations. Moreover, both $M_{sat}$ and the calculated concentration of Fe antisite defects, when rescaled to the experimental transition temperatures, increase in a similar

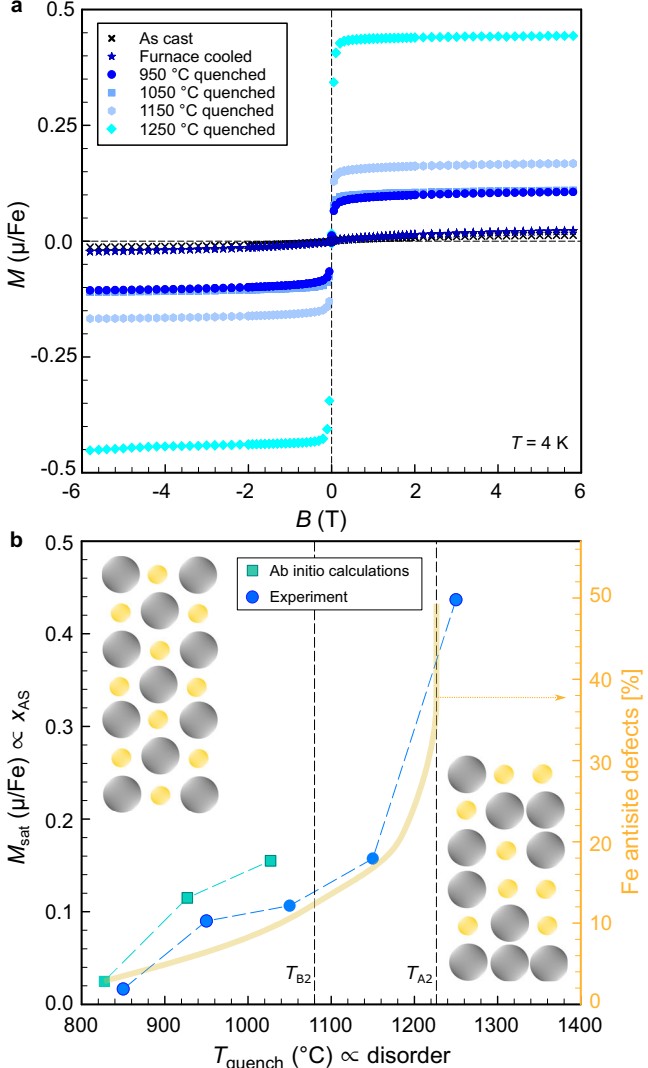

**Fig. 3 Quantification of disorder by low-temperature magnetisation.**
**a** Field-dependent magnetisation of $Fe_2VAl$ at $T = 4$ K for different quenching temperatures. **b** Experimental and calculated saturation magnetisation versus quenching temperature. The insets show a sketch of atomic disorder, increasing with $T_{quench}$. Right scale shows the calculated concentration of Fe antisite defects (yellow curve) from Fig. 1b, rescaled to the experimental transition temperatures $T_{B2}$, $T_{A2}$[22].

fashion showing an abrupt increase near $T_{A2}$. This demonstrates that the rapid quenching method could successfully introduce the magnetic antisites in these samples, which agrees well with our ab initio calculations.

**Thermoelectric properties.** Figure 4 shows the anomalous behaviour of the low-temperature electrical resistivity, thermopower and Hall mobility of as-cast and furnace-cooled $Fe_2VAl$ as well as the temperature-dependent electrical resistivity $\rho(T)$ and thermopower $S(T)$ of $Fe_2VAl$, measured in a wide temperature range from 4 to 800 K for all samples.

As-cast and furnace-cooled samples, according to the respective magnetic measurements, represent the dilute limit of antisites, where $E_F$ is expected to be situated within the narrow impurity band. In the theory of Mott and Anderson, electrons remain localised up to a critical concentration of impurities due to the combined effects of Mott and Anderson localisation[14,27]. At sufficiently low temperatures, the conduction via hopping

between localised impurity states, which are far apart in space but close in energy is more likely than nearest-neighbour hopping, leading to a characteristic temperature dependence. The low-temperature resistivity of our as-cast and furnace-cooled samples (see Fig. 4a) can be well described by such a phonon-assisted variable-range hopping (VRH) conduction[28]:

$$\rho(T) \propto \exp\left[\left(\frac{T_0}{T}\right)^{1/(d+1)}\right], \quad (1)$$

which specifies to $\rho(T) \propto \exp\left[\left(\frac{T_0}{T}\right)^{1/4}\right]$ in 3D, where $T_0$ is the characteristic Mott temperature. $T_0$ inversely depends on the localisation length $\xi_L = \left(\frac{1}{18}N(E_F)k_BT_0\right)^{-1/3}$, which diverges at the insulator–metal transition[29]; $N(E_F)$ being the DOS at the Fermi energy. The fitted values of $T_0$ are about 2−4 mK, which are at least five orders of magnitude lower than for VRH between localised donor and acceptor states in marginally doped semiconductors[30,31]. This suggests the presence of Anderson-localised states near $E_F$, similar to those reported in refs. [29,31–33] with large values of the localisation length in the order of $\xi_L \approx 1000$ Å, indicating direct proximity to the insulator–metal transition.

We note that a soft Coulomb gap in the presence of strong electronic interaction results in Efros–Shklovskii hopping[23], which would give a temperature dependence similar to Eq. (1), with the factor $T^{-1/(d+1)}$ being modified to $T^{-1/2}$ for all dimensions. However, the 3D variable range hopping gave the best fit to our data ($r^2 = 0.996$) compared to other temperature dependencies which we considered (see Supplementary Fig. 11a). Furthermore, the low-temperature behaviour of the thermopower significantly deviates from simple linear diffusion behaviour (see Supplementary Fig. 11b) and instead follows $S(T) \propto T^{1/2}$ (see Fig. 4b) which is also consistent with VRH in 3D[34], corroborating the resistivity data. Finally, the low-temperature Hall mobility $\mu_H(T)$ also shows an almost constant temperature dependence, with slightly increasing values (see Fig. 4c), consistent with localisation of charge carriers near $E_F$[35]. This picture of Anderson-localised states close to the Fermi level also reconciles many other peculiar properties of this compound, e.g., metallic thermodynamic and photoemission data in spite of semiconductor-like transport properties[36], negative magnetoresistance[37], etc., which have been an ongoing discussion over the past three decades[20,36,38].

Figure 4d shows $\rho(T)$ in a wide temperature range from 4 to 800 K for disorder-tuned stoichiometric $Fe_2VAl$ quenched at different temperatures. Above $T = 400−500$ K, a semiconductor-like behaviour of the resistivity, $d\rho/dT < 0$, is found for all samples. This can be attributed to the intrinsic pseudogap of the compound[36,39,40]. At lower temperatures, the behaviour gradually modifies from semiconductor-like to metallic with increasing antisite disorder, $d\rho/dT > 0$, in line with the picture of a continuous Anderson-type insulator–metal transition. The residual resistivity $\rho_0$ decreases by one order of magnitude with increasing $T_{quench}$, which also manifests itself by a substantial increase of the Hall carrier concentration (see Supplementary Fig. 8a). Furthermore, the appearance of metallic transport goes hand in hand with the development of a local maximum in $\rho(T)$ at a temperature $T_{\rho,max}$, which shifts to higher temperatures as $T_{quench}$ increases.

In Fig. 4e, $S(T)$ is shown from 4 to 800 K. As-cast and furnace-cooled samples display positive values of $S(T)$ and a pronounced maximum at ≈200 K, consistent with the narrow pseudogap band structure, where $E_F$ is situated near the valence band edge. As $T_{quench}$ and the antisite concentration increase, $S(T)$ becomes consistently smaller and even exhibits a sign reversal for $T_{quench} = 1000−1050$ °C over the whole temperature range. This implies a substantial contribution of the antisite electrons to $S(T)$

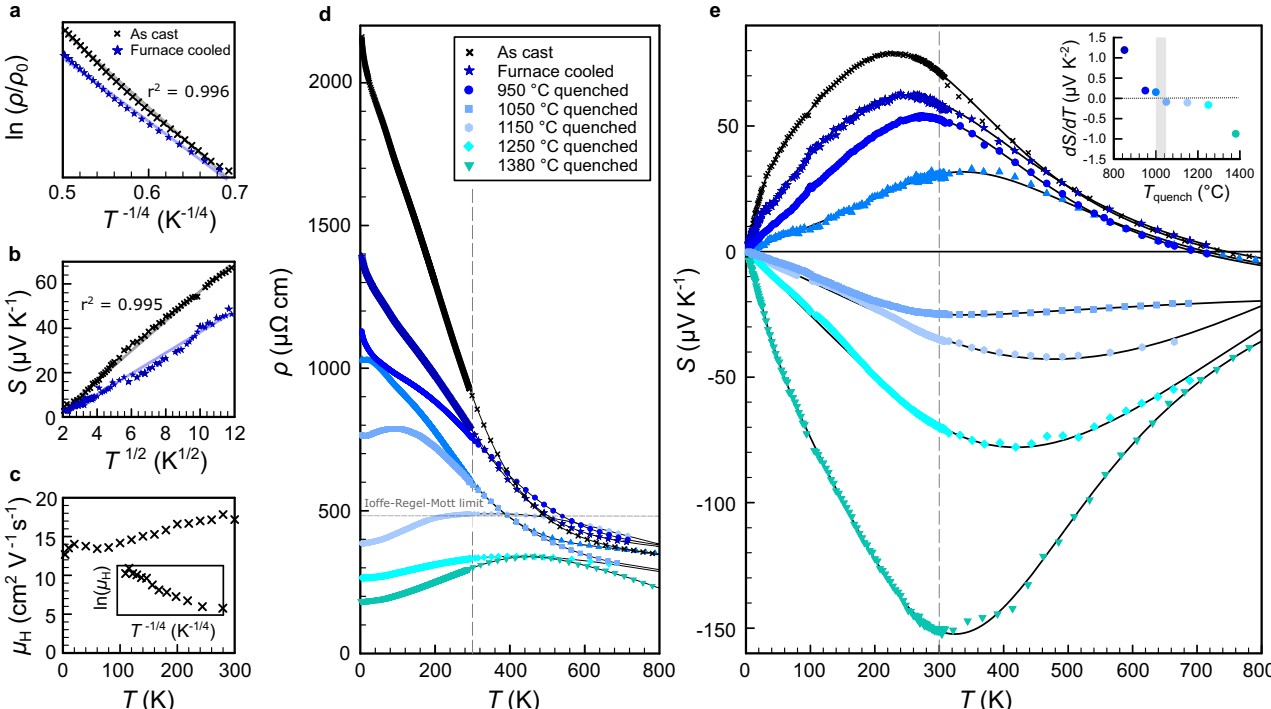

**Fig. 4 Thermoelectric transport across the Anderson localisation−delocalisation transition. a−c** Evidence from electrical resistivity, thermopower and Hall effect measurements for charge localisation and variable-range hopping behaviour in as-cast and furnace-cooled Fe$_2$VAl at low temperatures. **d** Temperature-dependent resistivity of Fe$_2$VAl with increasing quenching temperature. Dashed line denotes the Ioffe-Regel-Mott limit in Fe$_2$VAl. **e** Sign reversal of the temperature-dependent thermopower with increasing quenching temperature. Inset shows the slope of $S(T)$ at low temperatures for the different samples. Solid lines are guides to the eye.

in order to account for the dramatic change of the thermopower, from large p-type to large n-type values. It is remarkable that $S(T)$ becomes negative over the entire temperature range down to 4 K, which implicates a change in the electronic structure exactly at the Fermi level. Analysing the carrier-concentration-dependent thermopower (Pisarenko plot, Fig. 5b) suggests that electronic states are piled up in a narrow energy interval with the Fermi energy remaining inside the pseudogap, in agreement with density functional theory calculations (Supplementary Fig. 3). It is thus unlikely that the sign reversal of the thermopower results from a shift of the Fermi level into the conduction band.

## Discussion

Figure 5 shows in detail the evolution of the electronic transport across the Anderson transition and the thermoelectric performance of disorder-tuned, stoichiometric Fe$_2$VAl in comparison to conventional doping studies.

To gain further insight into the details of the electronic transport, we measured the Hall effect for several samples. In Fig. 5a, the Hall mobility $\mu_H$ is plotted as a function of the carrier concentration $n_H$. Usually, an increase of the carrier concentration simultaneously results in a decrease of carrier mobility due to additional carrier scattering. However, in Fe$_2$VAl the Hall mobility initially shows a steep rise upon a small increase of the carrier concentration. Such anomalous behaviour was also found near the Anderson-Mott transition of phosphorus-doped Si[41,42]. The Hall mobility initially increases by an order of magnitude when increasing the phosphorus concentration $n$ until the impurity band becomes delocalised, ultimately followed by a decrease of the mobility $\mu_H \propto 1/n$ in the metallic regime. This is indeed what is observed here as well. Our Hall data analysis suggests that at a critical quenching temperature $T^*_{quench} = 1000-1150°C$ the critical concentration of antisites, required for

delocalisation of the impurity band, is reached. We note that we find a very similar trend by analysing the data of a previous low-temperature magnetotransport study on Fe$_2$VAl$_{1-x}$Si$_x$[43], where $E_F$ is shifted from the localised impurity states into the extended states by extrinsic Si doping.

Figure 5b compares the carrier concentration dependence of the thermopower $S(n)$ for disorder-tuned Fe$_2$VAl and conventionally doped Fe$_2$VAl$_{1-x}$Si$_x$ Heusler compounds. In the case of Al/Si substitution, the electronic structure around $E_F$ is unaltered but $E_F$ is shifted into the conduction band (rigid-band doping). We find that the thermopower peaks at a doping concentration of $n_H \lesssim 10^{21}$ cm$^{-3}$ where the Fermi level is optimally placed with respect to the conduction band edge. Both the optimum doping concentration as well as the overall $S(n)$ behaviour, however, are in stark contrast to the results we get for disorder-tuned stoichiometric Fe$_2$VAl. Here, the thermopower steadily increases up to much higher values of $n_H$. Such a tendency indicates that electronic states are being piled up in a narrow energy interval. Thus, both the carrier concentration and the thermopower can increase simultaneously, which is beneficial for high thermoelectric performance.

In Fig. 5c, we show the peak values of the thermopower $S_{max}$ as well as the temperature of the maximum of $\rho(T)$, $T_{\rho,max}$, as a function of the quenching temperature. Again around $T^*_{quench}$, the thermopower displays a sign reversal and simultaneously $T_{\rho,max} > 0$ for the first time. According to the well-known modified Mott formula[44] the sign of the thermopower depends on the sign of the energy derivative of the density of states at the Fermi energy:

$$S \propto \left( -\frac{1}{N(E)} \frac{\partial N(E)}{\partial E} \right)_{E=E_F} \tag{2}$$

Since we intrinsically tuned the number of antisite defects within a single Fe$_2$VAl sample without extrinsic doping, the

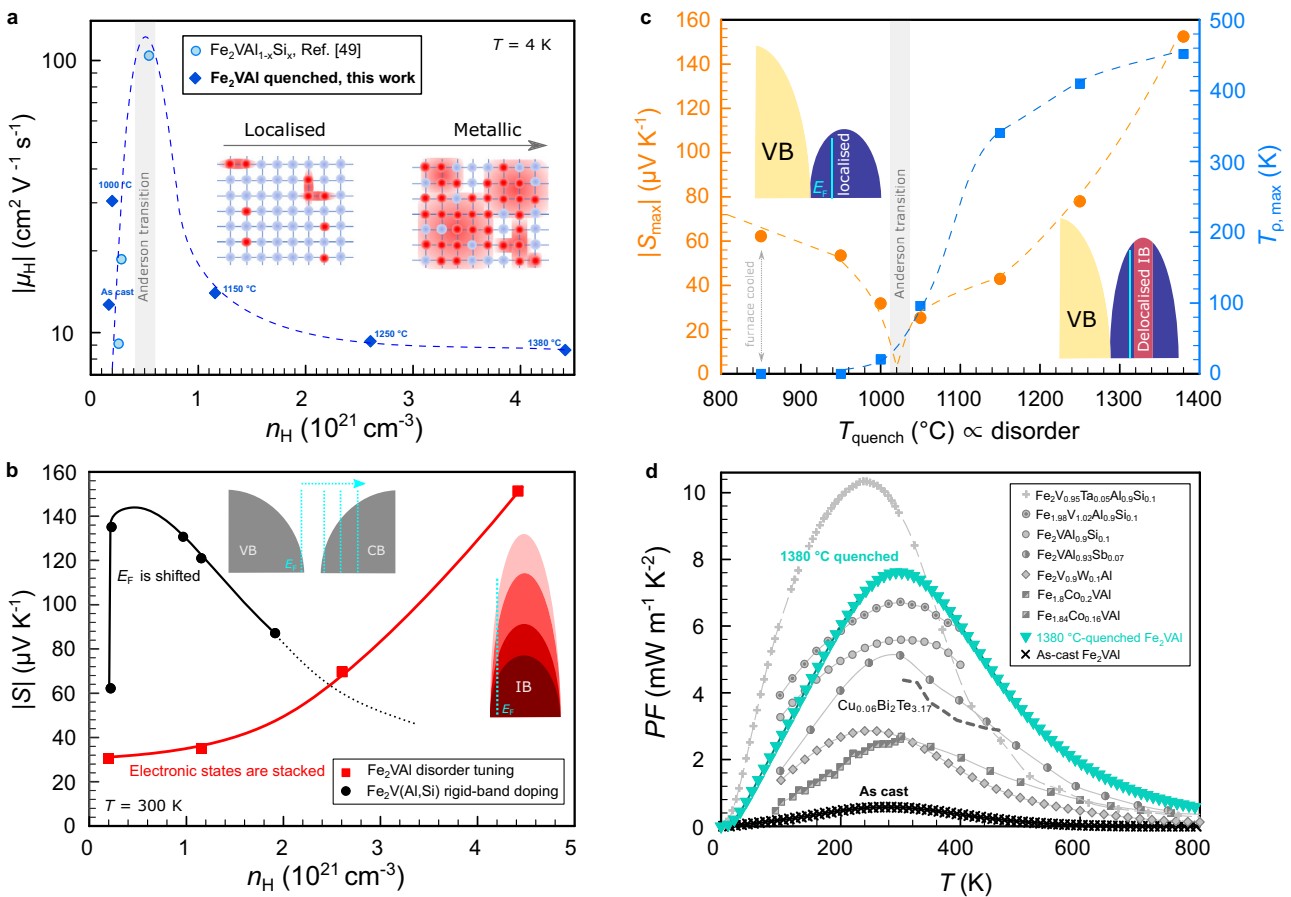

**Fig. 5 Delocalisation of impurity band and high thermoelectric performance driven by antisite disorder. a** Hall mobility versus carrier concentration at $T = 4$ K. **b** Thermopower versus carrier concentration (Pisarenko plot) for disorder-tuned Fe$_2$VAl and conventionally doped Fe$_2$VAl$_{1-x}$Si$_x$. **c** Peak values of the thermopower and temperature of the maximum of the resistivity versus quenching temperature. Dashed and solid lines are guides to the eye. **d** Power factor of as-cast and 1380 °C-quenched stoichiometric Fe$_2$VAl compared to the optimum power factors obtained by extrinsic n-type doping on the Al, V and Fe sites (Fe$_2$VAl$_{1-x}$Si$_x$[45], Fe$_2$VAl$_{1-x}$Sb$_x$[46], Fe$_2$V$_{1-x}$W$_x$Al[47], (Fe$_{1-x}$Co$_x$)$_2$VAl[48, 49]) as well as co-doping Fe$_2$V$_{0.95}$Ta$_{0.05}$Al$_{0.9}$Si$_{0.1}$[50] and off-stoichiometry Fe$_{1.98}$V$_{1.02}$Al$_{0.9}$Si$_{0.1}$[51]. For comparison, the highest n-type power factor reported for the state-of-the-art thermoelectric Bi$_2$Te$_3$-system[52] is plotted as dashed lines.

change in the sign of the thermopower can be directly related to the modification of the electronic structure close to $E_F$. Based on our DFT calculations shown in Fig. 2, it is evident that the sign reversal of $S(T)$ can be explained by the occurrence of sharp features in the DOS just above $E_F$ owing to the antisite defects, which is also supported by analysing the measured thermopower versus Hall carrier concentration in a Pisarenko plot as mentioned above (see Fig. 5b). However, initially these antisite states are Anderson-localised as indicated by the variable range hopping and by the carrier concentration dependence of the Hall mobility at low temperatures. Thus, they hardly contribute to the overall charge transport. However, when these states become delocalised at the Anderson transition, their colossal n-type contribution to $S(T)$ starts significantly exceeding the p-type contribution of the valence band resulting in large negative thermopower values $|S_{max}| > 150 \mu$V K$^{-1}$. Note, that the highest thermoelectric performance is not obtained directly at the critical concentration but at the metallic side of the Anderson transition since the impurity band needs to outweigh the contribution of the pristine material, which slightly modifies Mahan and Sofo's conditions for the best thermoelectric transport function (see Supplementary Fig. 12). The above-mentioned anomalies in thermoelectric transport, together with the Hall effect data, clearly indicate the delocalisation of the impurity band, as sketched in the insets of Fig. 5c. Further support for this argument is given by least-squares fitting

our temperature-dependent thermopower data to a charge transport model for delocalised impurity bands (see Supplementary Note 3). Our extended model indicates that a narrow impurity band, with a bandwidth $W = 0.03$ eV, yields the best agreement with our experimental data of the 1380 °C-quenched sample. However, $W$ goes to zero and the impurity band eventually becomes localised when the quenching temperature is decreased.

Figure 5d shows the power factor ($PF = S^2/\rho$) of as-cast and 1380 °C-quenched stoichiometric Fe$_2$VAl. We compare our results to various full-Heusler compounds that have been optimally doped by conventional substitution or off-stoichiometry and have been reported in the literature for their high power factors[45–51]. The maximum power factor of 1380 °C-quenched Fe$_2$VAl is 7.6 mW m$^{-1}$ K$^{-2}$, which is an enhancement by an order of magnitude compared to the as-cast sample and 30–40% higher than the best power factor for optimised rigid-band doping in this system[45] as well as among the highest power factors ever reported for any bulk material. Comparing to the highest n-type power factor of the state-of-the-art Bi$_2$Te$_3$-system[52] for example, $PF$ is about twice as high in our present study. The average power factor $PF_{ave}$ in the practical temperature range 293−503 K, where the majority of waste heat is released into the environment[53], reaches up to 6.2 mW m$^{-1}$ K$^{-2}$, higher than any other value reported previously for extrinsically doped samples.

It is noteworthy to mention that $\kappa_{ph}$ was also reduced by a factor of $2-3$ down to $\approx 10\,\mathrm{W\,m^{-1}\,K^{-1}}$ due to the disorder introduced by quenching (see Supplementary Fig. 8b). Consequently, this means that inducing disorder by thermal quenching is a new strategy that can enhance all thermoelectric properties at the same time, which is not achievable by conventional doping strategies.

In summary, we theoretically and experimentally demonstrated, how controlling the number of thermally activated antisite defects in a stoichiometric undoped Fe₂VAl bulk sample leads to an Anderson-type insulator–metal transition. This self-tuning mechanism of the disorder can significantly boost the thermoelectric performance within the very same sample by passing across the Anderson transition, where electrons are delocalised but occupy only a narrow interval in energy space. Indeed, we showed that Mahan and Sofo's 'best thermoelectric' is not just a mathematical construct but can be implemented in real materials by exploiting charge carriers on the metallic side of the Anderson transition in an impurity band. Moreover, controlling the level of disorder allows us to tune the optimal width of the energy-dependent transport function directly, which is not achievable by other band engineering strategies employed so far. Although disorder and charge localisation have been mostly considered as detrimental for thermoelectricity, our work discloses a novel paradigm to improve thermoelectric materials and devices via impurity conduction, employing temperature-induced disorder as a new tuning and control parameter.

## Methods

**Ab initio thermodynamic calculations.** Atomic ordering of bcc Fe₂VAl was studied by Monte Carlo simulations on the basis of the screened generalised perturbation method (SGPM)[54–56] with interactions obtained in the exact muffin-tin orbital coherent potential approximation (EMTO-CPA)[57,58]. The following configurational Hamiltonian has been used in statistical thermodynamics simulations

$$H = \frac{1}{2}\sum_p \sum_{\alpha,\beta\neq\delta} V_p^{(2)-\alpha\beta[\delta]} \sum_{i,j\in p} \delta c_i^\alpha \delta c_j^\beta$$
$$+ \frac{1}{3}\sum_t \sum_{\alpha,\beta,\gamma\neq\delta} V_t^{(3)-\alpha\beta\gamma[\delta]} \sum_{i,j,k} \delta c_i^\alpha \delta c_j^\beta \delta c_k^\gamma + \text{h.o.t.} \quad (3)$$

Here, the summation is performed over different types of clusters ($p$ and $t$ stand for indices of the pairs and triangles), alloy components (designated by Greek letters) and lattice sites ($i$, $j$ and $k$). $V_p^{(2)-\alpha\beta[\delta]}$ and $V_t^{(3)-\alpha\beta\gamma[\delta]}$ are the pair- and three-site effective interactions, which have been determined using the SGPM implemented in the Lyngby version of the EMTO-CPA code and $\delta c_i^\alpha = c_i^\alpha - c^\alpha$ is the concentration fluctuation of the $\alpha$ component from its average concentration $c^\alpha$ in the alloy at site $i$. The contribution from pair interactions in (1) can be reduced to a quasibinary form

$$H = -\frac{1}{2}\sum_p \sum_{\alpha\neq\beta} \tilde{V}_p^{(2)-\alpha\beta} \sum_{i,j\in p} \delta c_i^\alpha \delta c_j^\beta, \quad (4)$$

where $\tilde{V}_p^{(2)-\alpha\beta}$ are the usual binary effective interactions describing the mutual ordering of $\alpha$ and $\beta$ atoms and are related to the multicomponent effective pair interactions $V_p^{(2)-\alpha\beta[\delta]}$. In order to model the B2 and A2 order–disorder transitions, which both happen at high temperatures, the SGPM effective interactions have been calculated in a random Fe₀.₅V₀.₂₅Al₀.₂₅ alloy using a lattice parameter of $a = 2.998\,\text{Å}$, which roughly corresponds to the experimental one at the A2–B2 ordering transition. The one-electron excitations have been included using the Fermi-Dirac distribution function at 1500 K. Magnetic excitations of Fe and V atoms at 1500 K were modelled using the disordered local moment (DLM) model combined with a model, which takes into considerations longitudinal spin fluctuations (LSF). The DLM-LSF contribution to the entropy has been defined as

$$S_i^{lsf} = d\,(m_i), \quad (5)$$

where $m_i$ is the local magnetic moment of the $i$th component in the DLM-LSF state and the parameter $d$ is element- and, in general, state-specific. In the case of V, $d = 3$, while in the case of Fe, $d = 2$ has been chosen for Fe in the random Fe₀.₅V₀.₂₅Al₀.₂₅ alloy and $d = 1$ for Fe on the V-Al sublattice in the partially ordered B2-Fe(V,Al). Fe exhibits strong localised magnetic character on the V-Al sublattice in contrast to when being at its own sublattice, where it becomes a weak itinerant magnet. The partially ordered B2-Fe(V,Al) alloy with the composition $(\mathrm{Fe}_{0.9}\mathrm{V}_{0.05}\mathrm{Al}_{0.05})(\mathrm{Fe}_{0.1}\mathrm{V}_{0.45}\mathrm{Al}_{0.45})$ has also been used in SGPM effective cluster calculations due to the fact that V-Al effective interactions strongly renormalise in

the partially ordered B2-alloy. Therefore, this effect should be taken into consideration to produce the correct $B2 - L2_1$ ordering temperature. Strain-induced contributions connected with local lattice relaxations caused by the atomic size mismatch of the alloy components are not included in the effective SGPM cluster interactions and would have to be obtained separately. Due to the complexity of the system, i.e., multiple alloy components and non-trivial magnetism, these contributions were neglected in the calculations, which led to an overestimation of the A2−B2 ordering transition. Supplementary Fig. 1a shows the effective pair interactions in the random bcc A2-alloy at 1500 K for the different pairs. The strongest interaction at the first coordination shell is given for the Fe-Al pair, which leads to the B2-type ordering at high temperatures and which drives the first A2−B2 phase transition. One can see that the nearest-neighbour Fe-V interaction is rather weak, but also of the ordering type. There are also several strong three- and four-site interactions in this alloy, which affect the A2−B2 transition temperature, shifting it by about 200 K but which do not qualitatively change the picture of ordering. The A2−B2 ordering phase transition calculated from these interactions is about 2100 K in the Monte Carlo simulations, which is about 600 K higher than the experimental one due to the previously mentioned neglected strain-induced interactions. If Monte Carlo simulations were performed with the bcc effective interactions (obtained in the random bcc A2 alloy), the second $B2 - L2_1$ ordering transition would be at 1160 K, which is lower than the experimental one that happens at 1350 K. However, Supplementary Fig. 1b shows that the V-Al effective pair interactions are substantially renormalised at the V-Al sublattice where this transition happens such that the then calculated transition temperature is about 1330 K, which is only 20 K below the experimental one.

**Electronic structure calculations.** Calculations on random and ordered Fe₂VAl alloys have been done using the coherent potential approximation (CPA)[59] and locally self-consistent Green's function (LSGF) technique[60,61], which accurately accounts for the local environment effects in random alloys. Both these techniques have been used within the EMTO method referenced here as EMTO-CPA[62] and ELSGF[63], respectively. The EMTO-CPA calculations have been done with the Lyngby version of Green's function EMTO code, where the screened Coulomb interactions were calculated in the single-site DFT-CPA[64] and SGPM. The EMTO-CPA method was used for calculating the density of states (DOS) of the infinitely dilute limit of antisite disorder ($x_{AS} \to 0$) without taking into consideration the perturbation of the electronic structure of the nearest-neighbour atoms. While the results for the Fe$_V$ and V$_{Fe}$ antisite defects are shown in the main article the Fe$_{Al}$ defects show similar features (see Supplementary Fig. 2), namely a magnetic ground-state solution as well as localised electronic states near the Fermi level $E_F$.

Furthermore, to confirm the effect of increasing the antisite defect concentration in Fe₂VAl, supercells consisting of 108 atoms (54 Fe, 27 V, 27 Al) were created and the spin-polarised electronic structure was calculated using the Vienna Ab Initio Simulation Package (VASP)[65] as shown in Supplementary Fig. 3. For the supercell calculations, we used the standard general gradient approximation by Perdew, Burke, Ernzerhof (GGA-PBE)[66] for the exchange correlation term. After structural relaxation using a $\Gamma$-centred $3 \times 3 \times 3$ k-point mesh and a cutoff energy of 450 eV, we used a $5 \times 5 \times 5$ k-point mesh for the spin-polarised density of states calculations to attain high accuracy. Supplementary Fig. 3a–c clearly shows the occurrence of sharp features in the energy-dependent DOS inside the gap near $E_F$, reminding of localised states. These states become broader upon increasing the impurity concentration, eventually leading to the formation of new bands, which fill out the gap and turn the system more metallic-like. However, due to the periodic boundary conditions (Bloch's theorem) imposed on the system by the supercell approach the Anderson-localised nature of these states cannot be described by this method and will not reproduce the decoherence of the wave functions, i.e., their exponential decay. More advanced, cumbersome methods are required for predicting the correct delocalisation transition and electrical conductivity of such systems[25].

**Magnetic calculations.** The magnetic moment of Fe antisites on the V and Al sublattice was calculated to about 2.2 $\mu_B$ and 2.7 $\mu_B$, respectively, while the magnetic moment of V on the Fe sublattice was calculated as 0.9 $\mu_B$. We found that these results are very consistent in both the ferromagnetic and DLM calculations. Supplementary Fig. 4 shows the calculated magnetisation from the Monte Carlo-generated 1024 atoms supercells at high temperatures compared with the experimental saturation magnetisation. A remarkable agreement is found with respect to the experimental data supporting the Monte Carlo-simulated degree of disorder in Fe₂VAl as well as the level of disorder in the quenched samples.

**Experimental preparation of disorder-tuned samples.** Highly pure bulk elements (Fe 99.99%, V 99.93%, Al 99.999%) were stoichiometrically weighed and melted using a high-frequency induction heating technique. The ingots were melted several times to ensure homogeneity and the relative mass loss $\frac{\delta m}{m}$ after melting was extremely low (<0.04 %) such that the polycrystalline samples could be considered of utmost stoichiometric quality. After melting the as-cast Fe₂VAl ingot ($m \approx 6$ g) was evacuated in a quartz tube at $\approx 10^{-5}$ mbar and annealed at 1123 K for 168 h, followed by furnace cooling. The ingot was then cut into five rectangular pieces ($m \approx 0.05-0.15$ g) using an aluminium oxide cutting wheel. Sample #1 was

measured after furnace cooling (Labelled as 'Furnace cooled' in the article), while sample #2−#5 were subjected to further heat-treatment for 24 h at 1223, 1323, 1423 and 1523 K, respectively. This was followed by rapid quenching in cold water (samples are labelled as '950 °C, 1050 °C, 1150 °C, 1250 °C quenched' in the article). During this process the quartz ampoules containing the samples were backfilled with argon to ensure thermal conductance to the cold water bath. In order to verify the reproducibility of the dramatic change in the thermoelectric response, we prepared a second batch of samples, synthesised in the exact same manner. We cut a rectangular piece of the as-cast ingot (labelled as 'As cast' in the article ≙ sample #0), which was then used for measurements and annealed the other remaining part of the ingot at 1123 K for 168 h as for the first batch. Again, several pieces were cut from the ingot and subjected to further heat treatment for 24 h at 1273 K and 1653 K, respectively, followed by rapid quenching in water (labelled as '1000 °C, 1380 °C quenched' in the article). Due to the fact that the measured properties of these samples were perfectly consistent with the tendency of the other samples we concluded that our sample preparation set-up is consistent and reproducible.

**Sample characterisation**. We used high-resolution powder X-ray diffraction (XRD) to investigate the crystal structure. Samples were ground to a fine powder and probed with conventional Cu-K$\alpha$ radiation in a Bragg–Brentano ($\theta, \theta$)-geometry using a PANalytical XPert Pro MPD at the X-Ray Center, TU Wien. The room temperature XRD patterns shown in Supplementary Fig. 6a display no signs of any impurity phases and feature almost all peaks of the full-Heusler structure pattern although the (111) peak at ≈27° is very weak in all samples (see Supplementary Fig. 6b). This has been attributed to B2-type disorder induced by hand grinding and other cold work effects by Maier et al.[22] as well as by Van der Rest et al.[67] and is present in all samples as we have reported previously for many other Fe$_2$VAl-based full-Heusler compounds[21,47,68]. Unfortunately, this means that the evolution of increasing B2-disorder with increasing quenching temperature is hardly observable in XRD powder patterns. Furthermore, as has also been pointed out by Van der Rest et al.[67] previously the similar structure factors of Fe and V atoms make the observation of increasing D0$_3$ disorder almost impossible as well. Nonetheless, a slight weakening of the (200) peaks at ≈31° as well as a slight peak broadening of the (422) peak at ≈82° for the high-temperature quenched samples already hint towards increasing disorder. This is also corroborated by a slight increase of the lattice parameter $a$, which has been extracted by performing Rietveld refinements on the XRD patterns using the program PowderCell, from $a = 5.763 \pm 0.002$ Å, for the as-cast and furnace-cooled sample, to $a = 5.772 \pm 0.003$ Å, for the high-temperature quenched samples. We want to emphasise that the metastable A2 structure (vanishing (111) and (200) peaks) occuring in Fe$_2$VAl above ≈1500 K, could not be stabilised during the quenching process, which has also been pointed out by Van der Rest[67]. This is probably due to the strong Fe-Al interactions that lead to a quick B2 ordering at high temperatures as indicated by our simulations. Summarising, the X-ray diffraction techniques present only minor qualitative evidence for the amount of disorder in Fe$_2$VAl samples. While neutron diffraction is better for differentiating Fe and V scattering factors, Rietveld refinements are tricky and often not unambiguous due to the large amount of possibilities for the site occupancies in this ternary system. Therefore, we chose a combination of advanced statistical thermodynamics calculations and detailed measurements of the magnetic properties to effectively track the disorder in high-temperature quenched Fe$_2$VAl.

We probed the microstructure of our samples with a scanning electron microscope (Quanta 250 FEG) using a back-scattered electron (BSE) detector and checked the composition by means of energy dispersive X-ray (EDX) analysis. These measurements were performed at the University Service center for Transmission Electron Microscopy (USTEM). The SEM images displayed no signs of a secondary phase precipitation (see Supplementary Figs. 6c-d) that could have led to the change in magnetic and transport properties shown in the main article. The chemical composition of the different samples was also identical within the error bar of EDX measurements, which allowed us to confirm that a change in the stoichiometry did not occur during heat treatment. Therefore, we conclude that the temperature-induced antisite disorder must have caused the dramatic changes in physical properties, presented in the main article.

**Property measurements**. Temperature- and field-dependent measurements of the DC magnetisation were carried out on a CRYOGENIC superconducting quantum interference device (SQUID) in a temperature range from 3 K up to room temperature and field range from 0 up to 6 T. The isothermal magnetisation from 0 up to 6 T was measured at various temperatures. Supplementary Figs. 7a-d show the isothermal magnetisation of furnace-cooled and 1050 °C-quenched Fe$_2$VAl as well as the corresponding Arrot plots $M^2$ vs $B/M$, which show a strong curvature toward the $B/M$-axis precluding a ferromagnetic order transition in these samples.

The Hall resistance was measured with an in-house set-up using a He-cryostat and a 9 T superconducting magnet. The Van-der-Pauw method was used for spot-welding thin gold wires onto thin sample pieces with the appropriate geometry. The magnetic field was swept from −9 to 9 T at various temperatures from 4 K up to 300 K. For the as-cast and furnace-cooled samples, the anomalous contribution to the Hall effect was small and only relevant at low temperatures and low magnetic fields. We could thus easily extract the normal Hall coefficient from the slope of the

linear Hall resistance $R_0 = \frac{R_{xy}}{B}$ at higher fields where the anomalous contribution $4\pi R_S$ is saturated. The carrier concentration and carrier mobility of the dominant charge carrier were then evaluated by $\mu_H \equiv \frac{R_0}{R_{xx}}$ and $n_H \equiv \frac{1}{eR_0}$. For the high-temperature-quenched sample, which showed an anomalous Hall effect (AHE) over the whole measured temperature range, we expect an influence of the AHE at higher temperatures. One has to be aware that despite the linear Hall effect, there should exist both holes and electrons in this compound, which can make it difficult to interpret $n_H(T)$ and $\mu_H(T)$. However, at low temperatures, before $S(T)$ shows its pronounced maximum, the contribution from the dominant charge carrier should dominate the temperature-dependent behaviour. Supplementary Fig. 9a shows the temperature-dependent Hall carrier concentration obtained from our Hall effect measurements. It can be seen that the furnace-cooled and as-cast sample display almost identical behaviour, which is also reflected in the transport measurements shown in the main article. The disordered 1250 °C-quenched sample on the other hand has a carrier concentration which is about an order of magnitude larger and comparable to that of ordered Fe$_2$VAl$_{0.9}$Si$_{0.1}$. As explained in the main article, this can be understood from delocalisation of charge carriers in the impurity band marked by the appearance of two mobility edges.

The electrical resistivity at low temperatures from 4 K to 300 K was measured in an in-house He-cryostat using a four-probe method with thin gold wires spot-welded onto the sample surface. Above room temperature, the electrical resistivity was again measured by the four-probe method in a commercial set-up (ZEM3 by ADVANCE RIKO). The sample dimensions were measured with an approximate accuracy of 0.005−0.01 mm resulting in an error bar ≲2%.

The thermopower at low temperatures from 4 K to 300 K was measured in an in-house set-up using a toggled heating technique to cancel out spurious voltages. Chromel-constantan thermocouples were used and soldered onto copper wires, which were spot-welded at the ends of the sample. The results, while in very good agreement with the high-temperature data, were adapted by a constant factor to match the measured data of the commercial set-up (ZEM3 by ADVANCE RIKO) at $T \approx 300$ K.

The total thermal conductivity was measured by the laser flash method, which allows one to calculate the thermal diffusivity by measuring the time-dependent temperature signal. The specific heat was measured using a differential scanning calorimeter and the sample density was evaluated by making use of Archimedes' principle. The electronic contribution to the thermal conductivity was extracted from the Wiedemann-Franz law $\kappa_e/\sigma = L_0 T$ by assuming a constant Lorenz number $L_0 \approx 2.44 \times 10^{-8}$ W $\Omega$ K$^{-2}$. The thermal conductivity above room temperature for the as-cast and 1380 °C-quenched sample are shown in Supplementary Fig. 9b. A multifold reduction of the phonon contribution $\kappa_{ph}$ for the 1380 °C-quenched sample could be obtained. This can most likely be explained by the increased point defect scattering in the more disordered sample and further indicates drastic changes of not only the electronic but also the phononic structure due to the temperature-induced disorder. Additional studies on the beneficial effect of temperature-induced disorder to reduce the relatively large $\kappa_{ph}$ in this compound might be worthwile to be pursued.

**Reporting summary**. Further information on research design is available in the Nature Research Reporting Summary linked to this article.

## Data availability
The data that support the findings of this study are available from the corresponding authors upon reasonable request.

## Code availability
The computer codes that support the findings of this study are available from the corresponding authors upon reasonable request.

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

## Acknowledgements

Financial support for F.G., M.P., A.R., T.M. and E.B. came from the Japan Science and Technology Agency (JST), program MIRAI, JPMJMI19A1. Part of DFT simulations was performed on resources provided by the Swedish National Infrastructure for Computing (SNIC) at PDC (Stockholm) and NSC (Linköping) A.V.R. acknowledges a European Research Council grant, the VINNEX center Hero-m, financed by the Swedish Governmental Agency for Innovation Systems (VINNOVA), Swedish industry, and the Royal Institute of Technology (KTH). A.V.R. also gratefully acknowledges the financial support under the scope of the COMET program within the K2 Center "Integrated Computational

Material, Process and Product Engineering (IC-MPPE)" (Project No 859480). This program is supported by the Austrian Federal Ministries for Climate Action, Environment, Energy, Mobility, Innovation and Technology (BMK) and for Digital and Economic Affairs (BMDW), represented by the Austrian research funding association (FFG), and the federal states of Styria, Upper Austria and Tyrol. Moreover, Oleg Peil is thanked for insightful comments on the manuscript.

## Author contributions

S.K. and F.G. had the initial idea for the study. F.G. supervised and performed the measurements. S.K. gave the theoretical explanation. F.G., M.P. and A.P. wrote the initial draft. M.K. and M.P. contributed to the sample synthesis and measurements. A.V.R. carried out ab initio investigations of alloy thermodynamics and electronic structure calculations. F.G., M.P., A.R., A.V.R., S.K., M.R., H.M., A.P., T.M. and E.B. discussed the results and modified the manuscript. E.B. and T.M. organised the funding.

## Competing interests

The authors declare no competing interest.
