## [Peer Review File · Nature Communications]

Anderson transition in stoichiometric Fe₂VAl: High thermoelectric performance from impurity bandsThis manuscript has been previously reviewed at another journal that is not operating a transparent peer review scheme. This document only contains reviewer comments and rebuttal letters for versions considered at *Nature Communications*.

REVIEWER COMMENTS

Reviewer #1 (Remarks to the Author):

The authors have carefully addressed the comments, criticisms and suggestions of all three previous referee reports. Given the fact that this manuscript has already been seen by previous referees, the manuscript deserves a decision by the referees. My recommendation is to publish the manuscript in its present form.

Reviewer #2 (Remarks to the Author):

The authors study the impact of disorder on thermoelectric properties induced by varying quenching temperatures in Fe₂VAl. The results presented can be interesting and can be significance for the field of thermoelectrics moving forward. The reviewers have addressed some of the concerns in previous round of reviewer responses but also misinterpreted some of them. The manuscript should be considered for publication after revisions suggested below are addressed:

(1.) The authors say ".....usually optimisation of thermoelectric materials involves changing the position of the Fermi level, i.e., the total number of electrons, while leaving the electronic structure unchanged, so-called rigid-band doping. Despite being undoped" . Stressing on "despite being undoped" would have been appropriate for a study which does so in a rigid band-structure scenario. However, in the present case where the band-structure changes significantly near the Fermi-level with disordering, stressing on improved thermoelectric properties despite "an unchanged Fermi-level" position is not meaningful. Obviously, the changes in thermoelectric properties here are contributed by the emerging transport edge of the new impurity band.

(2.) I agree that changing the Fermi-level further by conventional doping may change the properties of this material. But how do the authors argue to change the Fermi-level significantly? The carrier concentration of the best sample (1380C-quenched sample) is metallic $> 4 \times 10^{21}$ cm⁻³. Conventional doping cannot be expected to further change this large concentration of carriers (and Fermi-level) very much and corresponding changes in properties cannot be expected to be significant. In other words, the the larger the density-of-states at the Fermi-level, the harder it is to change the Fermi-level. To my understanding this just suggests that the disordering effect has a much larger impact on thermoelectric properties than conventional doping procedure. The authors should make this point instead of stressing too much on the future prospects of conventional doping in the disordered material.

(3.) The authors show an impressive contrast in Figure 10b of the effect of disordering and conventional doping in rigid band structure. Given that this manuscript discusses a different and simple strategy in thermoelectrics, the authors should consider moving this Figure (and similar

Figure for power factor perhaps) in the main-text of the manuscript to give some context to the materials scientists among the readers.

Response to referees

Firstly, the authors want to thank both referees for their positive feedback and critical evaluation of our manuscript. In the new version of our manuscript, changes and additional text have been added in **red color**. In the following, point-by-point answers to the comments by referee #2 are in **blue color**.

Reviewer #2:

- (1) The authors say ".....usually optimisation of thermoelectric materials involves changing the position of the Fermi level, i.e., the total number of electrons, while leaving the electronic structure unchanged, so-called rigid-band doping. Despite being undoped" . Stressing on "despite being undoped" would have been appropriate for a study which does so in a rigid band-structure scenario. However, in the present case where the band-structure changes significantly near the Fermi-level with disordering, stressing on improved thermoelectric properties despite "an unchanged Fermi-level" position is not meaningful. Obviously, the changes in thermoelectric properties here are contributed by the emerging transport edge of the new impurity band.

The authors gratefully acknowledge the valuable criticism by the referee. Indeed, the referee makes a valid point here. Therefore, we have removed this sentence in the updated version of our manuscript.

- (2) I agree that changing the Fermi-level further by conventional doping may change the properties of this material. But how do the authors argue to change the Fermi-level significantly? The carrier concentration of the best sample (1380C-quenched sample) is metallic $> 4 \times 10^{21} \text{ cm}^{-3}$. Conventional doping cannot be expected to further change this large concentration of carriers (and Fermi-level) very much and corresponding changes in properties cannot be expected to be significant. In other words, the the larger the density-of-states at the Fermi-level, the harder it is to change the Fermi-level. To my understanding this just suggests that the disordering effect has a much larger impact on thermoelectric properties than conventional doping procedure. The authors should make this point instead of stressing too much on the future prospects of conventional doping in the disordered material.

It is true that the larger DOS at the Fermi energy can impose difficulties on shifting the position of the chemical potential. As suggested by the referee, we have focused our discussion on the effect of disorder and its inherently different behavior compared to conventional doping in the present version of the manuscript.

- (3) The authors show an impressive contrast in Figure 10b of the effect of disordering and conventional doping in rigid band structure. Given that this manuscript discusses a different and simple strategy in thermoelectrics, the authors should consider moving this Figure (and similar Figure for power factor perhaps) in the main-text of the manuscript to give some context to the materials scientists among the readers.

As suggested by the referee, we have put Fig. S10b from the previous version of our manuscript into our new version (see Fig.5b in the present manuscript) to give some context for the materials scientists among the readers and show the stark contrast to rigid-band doping.